# Approach to Cohort-Wide Re-Analysis of Exome Data in 1000 Individuals with Neurodevelopmental Disorders

**DOI:** 10.3390/genes14010030

**Published:** 2022-12-22

**Authors:** Insa Halfmeyer, Tobias Bartolomaeus, Bernt Popp, Maximilian Radtke, Tobias Helms, Julia Hentschel, Denny Popp, Rami Abou Jamra

**Affiliations:** 1Institute of Human Genetics, University of Leipzig Medical Center, 04103 Leipzig, Germany; 2Center of Functional Genomics, Berlin Institute of Health at Charité, Universitätsmedizin Berlin, Hessische Straße 4A, 10115 Berlin, Germany; 3Limbus Medical Technologies GmbH, Neuer Markt 9/10, 18055 Rostock, Germany

**Keywords:** re-analysis, exome sequencing, neurodevelopmental disorder

## Abstract

The re-analysis of nondiagnostic exome sequencing (ES) has the potential to increase diagnostic yields in individuals with rare diseases, but its implementation in the daily routines of laboratories is limited due to restricted capacities. Here, we describe a systematic approach to re-analyse the ES data of a cohort consisting of 1040 diagnostic and nondiagnostic samples. We applied a strict filter cascade to reveal the most promising single-nucleotide variants (SNVs) of the whole cohort, which led to an average of 0.77 variants per individual that had to be manually evaluated. This variant set revealed seven novel diagnoses (0.8% of all nondiagnostic cases) and two secondary findings. Thirteen additional variants were identified by a scientific approach prior to this re-analysis and were also present in this variant set. This resulted in a total increase in the diagnostic yield of 2.3%. The filter cascade was optimised during the course of the study and finally resulted in sensitivity of 85%. After applying the filter cascade, our re-analysis took 20 h and enabled a workflow that can be used repeatedly. This work is intended to provide a practical recommendation for other laboratories wishing to introduce a resource-efficient re-analysis strategy into their clinical routine.

## 1. Introduction

Next-generation sequencing technologies have made it possible to bring approaches such as the cost-effective exome sequencing (ES) of individuals with a disease of presumed genetic origin into everyday clinical practice.

ES has a reported diagnostic yield of 30–50% in neurodevelopmental disorders (NDD) [1]. However, many affected individuals remain undiagnosed, which hinders appropriate clinical care. With increasing knowledge of gene–disease associations, the rising number of entries in variant databases [2], the implementation of functional studies and the improvement of bioinformatics tools, the re-analysis of nondiagnostic cases is one way to close this diagnostic gap [3].

The exact increase in the diagnostic yield obtained through re-analysis varies between studies, but was recently summarised in a review as 10% overall [4]. Typically, cohorts of approximately 50–100 individuals are manually re-analysed case by case [5,6,7]. Although this is readily conceivable in a research context, it is far from the reality of laboratories in a diagnostic setting, with limited staff capacity and a lack of reimbursement options. This raises the following question: how can we achieve the important task of re-analysis that can help to diagnose so many affected individuals? 

Others have already shown that, for larger cohorts, a semi-automated re-analysis is more appropriate and significantly reduces the workload per case, while still increasing the diagnostic yield [8,9]. Here, we describe a systematic approach for re-analysing ES data from a cohort of 1040 individuals. In addition to the increased diagnostic yield, we describe the sensitivity of the filter cascade and its adjustments so that this workflow can be performed repeatedly in a time-effective manner and maximises the clinical impact while maintaining a small burden on the laboratory. Finally, we highlight the pitfalls and develop recommendations to help other laboratories that have to balance day-to-day analyses and subsequent re-analyses.

## 2. Materials and Methods

We re-analysed individuals with severe, early-onset diseases, mainly with NDD (i.e., intellectual disability, epilepsy, autism) (979/1040, 94.1%, see Figure 1D and Appendix A). Expanding to other disease groups was beyond the scope of this work. We included individuals that had been analysed via ES over five years, between February 2017 and January 2022 (hereafter referred to as the “initial analysis”). The final cohort consisted of 1040 affected individuals from 983 families. 

In the initial analysis, we identified one or more (L)P disease-causing single-nucleotide variants (SNV) in 138/1040 individuals and (L)P copy-number variants (CNV) in 18/1040 individuals (hereafter referred to as “initially reported variants”, see Appendix A). This resulted in an ES diagnostic yield of 15% (155/1040, one individual had one SNV and one CNV). This number appears to be low; however, this is due to the multi-gene panel diagnostics, which clarified many cases before ES. 

Prior to this study, re-evaluation was requested in some cases by the referring clinician but did not lead to positive reports. Most Trio-Exomes were assessed in a research context and promising candidate variants were reported. In thirteen individuals, such a candidate was subsequently published (see Appendix A), making them valid results in a diagnostic setting.

In 159/1040 cases, the DNA was enriched using SureSelect Human All Exon V6 (Agilent Technologies, Santa Clara, CA, USA) (see Figure 1B). In 534/1040 cases, the BGI Exome Capture 59M Kit (BGI, Shenzhen, China) was performed, and in 347/1040 cases, the TWIST Human Core Exome Kit (TWIST Bioscience, San Francisco, CA, USA) was used for DNA enrichment. In 145/1040 cases, only the affected individual (“Single”) or the individual with one parent was sequenced (“Duo”), while 895/1040 received Trio- or Quattro-ES (see. Figure 1A). Initial analysis was done using Varvis^®^ genomics software (Limbus Medical Technologies GmbH, Rostock, Germany), using hg19 as a reference. 

Prior to the initial ES analysis, 699/1040 individuals received multi-gene panel diagnostics (e.g., TruSightOne, Illumina, 4813 genes) and/or arrays (see Figure 1C). The remaining 341 individuals were the nondiagnostic cases of a larger ES cohort that were subsequently assessed on a research basis. The ES research assessment was regarded as the “initial analysis”, since it was the analysis that we re-analysed in this study.

For this study, we reprocessed all cases using an updated bioinformatics pipeline. Sequencing data were aligned to the human genome hg38. Variants were called from the resulting bam files using the GATK HaplotypeCaller [10] (version 4.2.0.0) and SNVs were annotated using vsWarehouse (Golden Helix, Inc., Bozeman, MT, USA, www.goldenhelix.com) (for further details, see File S1). The median time for BAM realignment (including all conversion and GATK pipeline recommended steps, such as deduplication and base quality score recalibration) was 1.79 h per sample, and the median time for VCF calling was 0.15 h per sample (PowerEdge R7515 Server; CPU: AMD EPYC 7702P 2.00 GHz with 64C/128 T; RAM: 196 GB; disk: 1.6 TB NVMe + 70 TB RAID). For all included subjects, the genomic regions targeted by the respective enrichment design had an average coverage of 149 reads, and >98% were covered by at least 15 reads. Copy number variants based on NGS were excluded from this analysis.

We then applied a filter cascade to identify the most promising variants. The multistep cascade included general filtering steps: (a) only SNVs with reliable quality; (b) only SNVs in diagnostically relevant genes, i.e., in genes that have been associated with a phenotype (morbidgenes.org, monthly updated database; version v2022-03.1 was used in this study and contained 4772 genes); (c) only SNVs in genes with a sufficient phenotypic overlap with the individual symptoms based on HPO terms [11], using the HPOsim score (Limbus Medical Technologies GmbH, Rostock, Germany) at a threshold of 0.1. This score reflects the similarity of the HPO term set of the individual and of the corresponding gene (for further details, see File S1).

Subsequently, we filtered the remaining SNVs based on different inheritance modi (see Figure 2).

Autosomal dominant: All rare heterozygous variants (not in gnomAD [12], release 2.0.1) were evaluated if they were linked to an autosomal dominant mode of inheritance in OMIM [13], and (a) *de novo* missense with a high missense gene constraint (Z score) [14,15], (b) *de novo* missense predicted to be deleterious by at least 4/5 in silico prediction tools (SIFT, PolyPhen2, MutationTaster, MutationAssessor, and FATHMM), (c) predicted to result in a loss of function (LoF) with a high LoF gene constraint (pLI score) [12], or (d) had previously been reported in the ClinVar database [16] as “likely pathogenic” or “pathogenic” ((L)P) by a laboratory other than ours until 7 April 2022.

Autosomal recessive: All homozygous variants were evaluated if they were not found in a homozygous state in gnomAD, with a minor allele frequency (MAF) < 0.01% and (a) predicted to result in a LoF, or (b) reported as (L)P in ClinVar. For compound-heterozygous variants, we filtered for genes with at least two variants that had an MAF < 0.01% in gnomAD and that were both not found in a homozygous state in gnomAD. In addition, the gene had to be linked to an autosomal recessive inheritance in OMIM and one of the variants had to (a) result in an LoF, or (b) be reported as (L)P in ClinVar previously. The other variant did not need to result in an LoF or to be reported as (L)P in ClinVar previously.

X-linked: In male individuals, we filtered for hemizygous variants on the X chromosome that were not found in a hemizygous state in gnomAD, with an MAF < 0.001%, that were (a) *de novo*, (b) predicted to result in an LoF, or (c) have been reported as (L)P in ClinVar. X-linked, dominant disorders are covered by the autosomal dominant filter.

The variants that passed the above-mentioned filter cascade were exported to Excel software (Microsoft Corporation, Redmond, Washington, DC, USA), and manually evaluated one by one by an experienced geneticist. The evaluation included checking the phenotypic overlap of the respective individual and the variant-associated disorder using the OMIM database, HGMD [17], or via literature research in PubMed. The DECIPHER database [18] was used for the visualisation of mutational hotspots. If needed, variant quality was assessed using Integrative Genomics Viewer [19]. For splicing prediction, variants were further assessed using AlamutVisual (Interactive Biosoftware, Rouen, France, v2.7.2) and spliceAI [20]. Classification was done according to the ACMG criteria [21], the ACGS Best Practice Guidelines for Variant Classification [22], and the latest updates published by the ClinGen consortium (https://clinicalgenome.org/working-groups/sequence-variant-interpretation/, accessed on 19 September 2022). If needed, validation via Sanger or Nanopore Sequencing was performed (for further details, see File S1). If relevant variants were identified, the referring clinicians were contacted.

The study was conducted in accordance with the Declaration of Helsinki and approved by the Ethics Committee of University of Leipzig, Germany (224/16-ek and 402/16-ek). Written informed consent for genetic testing and the publication of findings after providing advice and information about the study was obtained from all study subjects or their legal representatives.

## 3. Results

### 3.1. Manual Evaluation of Filtered Variants 

We applied the described filter cascade (see Figure 2 and Methods) on the whole cohort, which revealed a list of 802 variants (0.77 variants per case). While some of the variants needed intensive research, others were easy to interpret. The manual evaluation of all variants took around 20 h. The heterozygous variants (see Figure 2) accounted for the largest proportion of all variants (493/802, 61.5%), followed by the compound heterozygous variants (212/802, 26.4%), while homozygous (73/802, 9.1%) and hemizygous (24/802, 3%) variants had smaller proportions. 

### 3.2. Novel Diagnoses through Re-Analysis

Re-analysis of all samples revealed an additional nine (L)P SNVs (hereafter referred to as “novel” variants, see Appendix A). Seven variants were disease-causing (primary findings), i.e., 0.8% (7 of 885 nondiagnostic exomes) diagnostic yield. Furthermore, two pathogenic variants were identified in two genes out of a list of 72 genes recommended to be reported as secondary findings by the ACMG guidelines (v3.0) [23]. Seven of these nine novel variants were heterozygous, one was hemizygous, and one was homozygous (see Table 1, and further details in File S1 and Appendix A). Of the nine novel variants, in five cases, the gene–disease association was not published at the time of the initial analysis (*FOXP4, LMNB1, MORC2, MSL3, SORD*). The median interval between the initial negative report and the publication of new data on morbidity was nine months (3–53 months). The variants in *KMT2C, MTOR,* and *SDHC* were initially not called with freebayes (v1.1.0-9-g09d4ecf) [24], whereas the GATK HaplotypeCaller (version 4.2.0.0) managed to detect these variants. Moreover, in one variant, the gene (*TTN*) was not on the ACMG secondary findings list at the time of the initial analysis and has only recently been added [23]. We submitted the nine novel variants to the ClinVar database and re-contacted all referring clinicians of affected individuals with newly identified (L)P variants and all requested an updated report.

### 3.3. Sensitivity and Filter Adjustments

Our re-analysis correctly captured 125 of 147 initially reported diagnostic SNVs among 138 individuals, which resulted in sensitivity of 85% (further details in Appendix A). Twenty-two initially reported (L)P SNVs could not be captured with our filter cascade, hereafter referred to as “lost” variants. Nine variants were too frequent in gnomAD (9/22, 41%) and six SNVs (6/22, 27%) were not detectable through our filter criteria (e.g., one homozygous missense variant that is not reported in ClinVar as (L)P; further details in Appendix A). Two LoF variants were found in genes with a pLI score < 0.9, and one missense variant was found in a gene with a low Z score, which were filtered out. In addition, two variants had low quality in the re-analysis and one variant was filtered out, as the mode of inheritance in the associated gene did not match the mode of inheritance in OMIM. One variant was lost due to two reasons: a low Z score and mismatch in the mode of inheritance in OMIM.

The Z score threshold for missense variants was optimised during the course of the study. At first, a Z score of 4.0 for *de novo* missense variants was set, which resulted in a workload of 60 heterozygous variants. Compared with the initially reported pathogenic SNVs, this led to the loss of 62% (32/52, sensitivity of 38% for this filter approach) of the initially reported de novo missense variants. If only considering the Z score as a filter criterion, we found that a cut-off of 2.2 only lost three initially reported variants while keeping a moderate workload of 180 variants (see Figure 3A). Out of these three, one was rescued by another filter criterion (reported (L)P in ClinVar).

Application of the HPOsim score filter with a threshold of 0.1 reduced the variant evaluation burden by 35% (1236 vs. 802 variants; notably, not all variants have an HPOSim score), while three initially reported variants were lost (see Figure 3B). Two of them only partially explained the individuals’ symptoms. In the third, the HPO terms used did not result in a sufficiently high overlap. In our opinion, the threshold of 0.1 provides the best balance between clinical sensitivity and workload.

## 4. Discussion

Here, we present an efficient strategy for the high-throughput re-analysis of ES data in severely affected individuals. This workflow is easily applicable to large cohorts and prioritises the identification of (L)P variants based on ACMG criteria. In our cohort of 1040 affected individuals, this approach resulted in nine clinically relevant findings, with an assessment time of 20 h for the entire cohort. We summarise our recommendations in Figure 4.

Our results confirm the general assumption that the re-analysis of older cases is worthwhile, due to updated bioinformatic approaches and protocols (e.g., caller and alignment), the increasing knowledge of gene–disease associations and the rising number of entries in variant databases [2]. In our re-analysis, we did not identify (L)P variants that were reportable at the time of the initial analysis. This means that we did not uncover any human shortcomings. Interestingly, in our re-analysis, five out of the nine novel variants were classified as pathogenic because new data on the morbidity of the gene were published in the meantime. The median interval between the initial report date and the studies indicating the morbidity of the gene was nine months. A re-analysis after 15 months would have been beneficial in four out of five cases. Considering the small number of cases, no exact time interval for the optimal initiation time of a re-analysis can be given based on our data. However, in line with other authors, we suggest re-analysing ES data 18–24 months after a negative report [4,25].

In our cohort, heterozygous variants accounted for the largest proportion of all manually evaluated variants and represent 7/9 novel variants. This is in accordance with reports demonstrating that autosomal dominant inherited disorders account for the largest proportion in NDD [26]. All novel reported heterozygous variants in primary findings occurred *de novo*, so we recommend focusing on this subgroup if the dataset allows. Notably, the compound heterozygous variants accounted for the second-largest proportion of all variants, while not leading to a novel diagnosis. The evaluation of 97 homozygous and hemizygous variants resulted in one novel diagnosis each (*MSL3* in hemizygous, *SORD* in homozygous variants). 

In 699/1040 individuals, the ES was performed following negative multi-gene panel diagnostics and/or array analysis. In the 341/1040 remaining cases, no other genetic testing was performed before ES. After ES remained negative in a diagnostic assessment, these 341 individuals were subsequently included and assessed in a research cohort. The initial diagnostic yield of 15% seems to be low. However, this is due to the cohort being preselected to include only individuals who were unremarkable in previous multi-gene panel diagnostics (e.g., TruSightOne, Illumina, 4813 genes) or in a pure diagnostic ES assessment. Additionally, all Trio-Exomes were analysed regarding new gene–disease associations and promising candidates were successively reported in a research setting. In thirteen individuals, such a candidate was published in the time between the initial analysis and this study (see Appendix A). As we perform research in parallel to diagnostics at our institute, we add such novel variants during routine operation. Institutions that perform solely diagnostics would only find these thirteen variants during a re-analysis. Thus, the corrected diagnostic yield through re-analysis is 2.3% (20 novel primary findings in 885 nondiagnostic individuals). 

The application of an updated analysis pipeline with an alignment against hg38 and new callers did not fail to detect initially reported variants. In contrast, three variants (in *KMT2C, MTOR,* and *SDHC*) were identified that were missed with the initially used caller and software (freebayes and Varvis^®^). The other six novel variants, as well as the thirteen research variants, could have been identified in the original data. Thus, taking into account the infeasibility of the application of novel bioinformatics pipelines, new alignments and variant calling can be omitted. A re-annotation with the latest information was sufficient to identify 17/20 variants. However, if one would consider the re-processing of the data (alignment, calling, and annotation), we recommend that this should be done only if significant modifications are introduced to an analysis pipeline, including the calling software (e.g., GATK) and the analysis software (in our case, Varvis^®^ vs. GoldenHelix, but other providers exist).

The diagnostic yield of large, cohort-wide re-analyses [8] is below that of case-level re-analyses [5,27]. Our corrected diagnostic yield of 2.3% is in line with a study using a similar approach in a cohort of 4411 individuals that recently reported a 2.7% increase in diagnostic yield [8]. However, this and other large, cohort-wide studies [8,9,28] re-analysed only nondiagnostic cases. The inclusion of diagnostic cases allowed us to examine the sensitivity of our filter cascade and to address recommendations on how to optimise them. 

With the filter cascade described above, we correctly detected 125/147 (85%) initially reported disease-causing variants in the 1040 individuals. Of the 147 initially reported SNVs, 64 were heterozygous missense changes (52 of them *de novo*, see Appendix A). The iteration of missense Z score thresholds (see Figure 3) in *de novo* missense variants led to a diagnostic optimum at 2.2, with sensitivity of 94 % (49/52), while maintaining a moderate workload (180 variants to be manually evaluated). 

Two of the 22 lost variants were not detected because they failed our gnomAD frequency filter for heterozygous variants (see Appendix A). Rescuing one of them by adding variants that are present only once in gnomAD is possible in this filtering step, but results in an additional workload of 86 heterozygous variants. To rescue both variants (including NM_145239.2: c.649dup in *PRRT2*), the needed MAF must be as high as 0.4%, which is not feasible.

The filtering step for likely LoF variants took into account a pLI score > 0.9. One variant in *COL10A1* (pLI: 0) and one in *MECP2* (pLI: 0.894) did not pass this filter and were also not rescued by other filter criteria. An additional eight variants (in *GH1, H1-4, MECP2, NARS1, NRP2, PPM1D, PRRT2*) also did not pass this filter but were rescued by other filter criteria. Thus, if only considering pLI values > 0.9 in genes with LoF variants, we lost ten of 34 initially reported (L)P variants in eight genes (sensitivity of 71% for this filter approach; see also Appendix A). Our filter cascade is partially based on the assumption that severe, early-onset diseases lead to reduced reproduction in affected individuals and that causative variants are thus subject to quantifiable selection pressure (MAF and pLI). The lost variants in *PRRT2* and *COL10A1* are good examples demonstrating that some conditions cannot be precisely distinguished from this group of severe disorders. Do the epileptic seizures of a 6-month-old infant occur in the context of developmental and epileptic encephalopathy, or in the context of *PRRT2*-associated benign epilepsy (OMIM #605751), in which seizures are confined to infancy [29]? While the former is usually associated with reduced reproduction (and high pLI), the latter tends to be unaffected and consequently does not lead to selection pressure. In conditions with reduced penetrance, as described for *COL10A1*-related metaphyseal chondrodysplasia (OMIM #156500), the causative variants are also subject to comparatively less selective pressure [30]. X-linked diseases are also not necessarily subject to the same selection pressure in gnomAD. In our re-analysis, two initially reported variants in *MECP2* (Rett syndrome, OMIM #312750) did not pass the pLI filter. We therefore recommend less stringent filters for X-linked variants in female individuals. The reasoning for the other five genes (*NARS1, PPM1D, GH1, NPR2, H1-4)* that we may have lost with our pLI threshold are further described in Appendix A. 

Our filtering cascade may be adapted to identify such variants in less severe or less penetrant diseases or in genes that slightly miss the pLI threshold, e.g., by a “white list” of genes with low pLI and LoF as a known pathomechanism (including the abovementioned genes as well as further genes, e.g., *BRCA1* and *BRCA2*). However, such adjustments go beyond the scope of this work, and as 31 of the 34 variants could be identified based on other steps of the filtering cascade, we decided not to adjust the pLI cut-off.

In four individuals, the initial single ES led to the segregation analysis of missense variants and confirmation of *de novo* occurrence. In another male individual, a maternal missense variant in the X-chromosomal gene *ABCD1* was reported as disease-causing. These five variants were not identified with our filter cascade (see Appendix A). Additionally, 4/9 of the novel reported variants are *de novo* missense as well. Unless analysed in a trio approach or in a comprehensive variant assessment at the case level, these missense variants (i.e., not in ClinVar, no segregation information, or X-chromosomal) cannot be identified in a stringent, cohort-wide filter cascade. This demonstrates that the trio approach is of lasting advantage, as it also benefits the subsequent re-analysis.

If no trio information is available, these variants will probably continue to be lost in the short term. In the long term, it is likely that further criteria of the ACMG classification, such as PM1 (variant is located in mutational hotspot or in functional domain) or PS1/PM5 (amino acid exchange at the same amino acid position) will be parameterised and become available for a filter cascade. 

Of the 147 (L)P initially reported variants, 22 are compound heterozygous. We lost six of them with our filter cascade (see Appendix A) because they are too common (MAF 0.01–0.1%). Raising the threshold for the MAF leads to a substantial increase in the number of variants to be evaluated manually, analogous to the heterozygous variants. Since the group of compound heterozygous variants is already the second largest in our filter cascade and no novel variants were identified, this subgroup is most negligible if resources are limited. However, it should be taken into account that autosomal recessive forms of NDD [31] are not as well clarified and that substantial development in this field is expected.

Application of the HPOsim score resulted in a substantial reduction in variants to be manually evaluated. Review of the HPOsim score curves (see Figure 3B) does not allow the recommendation of a clear threshold. By choosing a threshold of 0.1, we lost only three initially reported variants. These variants do not explain the entire phenotype and are likely to be lost in any phenotype-based filtering approach (see Appendix A). There are several algorithms to calculate the phenotypic overlap (e.g., Phenomizer [32]). If the phenotypic overlap of an individual’s presentation with a particular disease is not possible, we recommend a stricter assignment of individuals to a disease group (e.g., “focal epilepsy”). Thus, individuals of the same disease group can be re-analysed only with respect to a specific gene panel (e.g., by using SysNDD [33] or PanelApp [34]). This allows a faster evaluation of the cohort or improvement of the sensitivity due to less stringent filters (e.g., gnomAD < 10).

To enable the application of our approach to many other laboratories, we chose OMIM as the database for disorders and retrieved the inheritance modes of these disorders from there. Although a dominant mode of inheritance is described in the literature, two heterozygous variants did not pass our filter cascade because the associated disorders are deposited in OMIM only with a recessive mode of inheritance. While such weaknesses in the databases can hardly be compensated by the users, they are subject to the constant updates in the knowledge of genetic diseases and thus become more and more negligible.

## 5. Conclusions

As our results show, re-analysis is worthwhile and, most importantly, feasible for large cohorts. In this study, we highlighted pitfalls and provided recommendations based on our experience to facilitate application for other laboratories (see Figure 4). The cut-offs of the above filter cascade ultimately depend on the setting in which the re-analysis takes place. Certainly, the highest possible sensitivity is to be aimed for, but the workload must correspond to the resources of the laboratory. 

We recommend a recall system in order to easily queue cases and perform re-analysis on a regular basis. If such systems are not available, software developers should be encouraged to implement recall algorithms, as re-analysis is both feasible and necessary. Smart gene- and disorder-specific filtering would allow us to set less stringent cut-offs regarding in silico predictions, MAF, etc., without increasing the workload, thus making the re-analysis more sensitive and more specific. This would also allow the expansion of the re-analysis to phenotypes that are not severe or of early onset.

Although there were edge cases that were presumably missed with our filter cascade, we conclude from our results that the filters stated above are a reasonable compromise between novel identified variants and needed workforce. In our laboratory, we plan to re-annotate all NGS data and use these filters on a semi-annual basis for cases that have not been analysed in the last 18 months.

## Figures and Tables

**Figure 1 genes-14-00030-f001:**
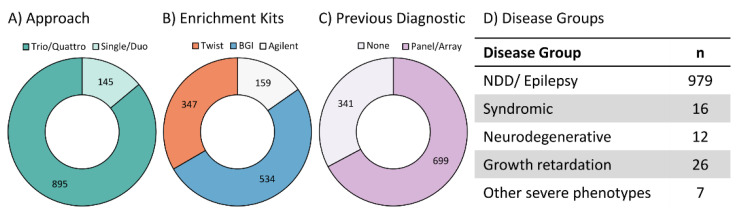
Cohort structure. Cohort structure of 1040 individuals regarding sequencing approach (**A**); enrichment kits used for ES—Agilent SureSelect Human All Exon V6, BGI Exome capture 59 M kit or TWIST Human Core Exome Kit (**B**); genetic tests conducted prior to exome sequencing (**C**); and disorder group (**D**). Numbers refer to the number of individuals. NDD: neurodevelopmental disorders.

**Figure 2 genes-14-00030-f002:**
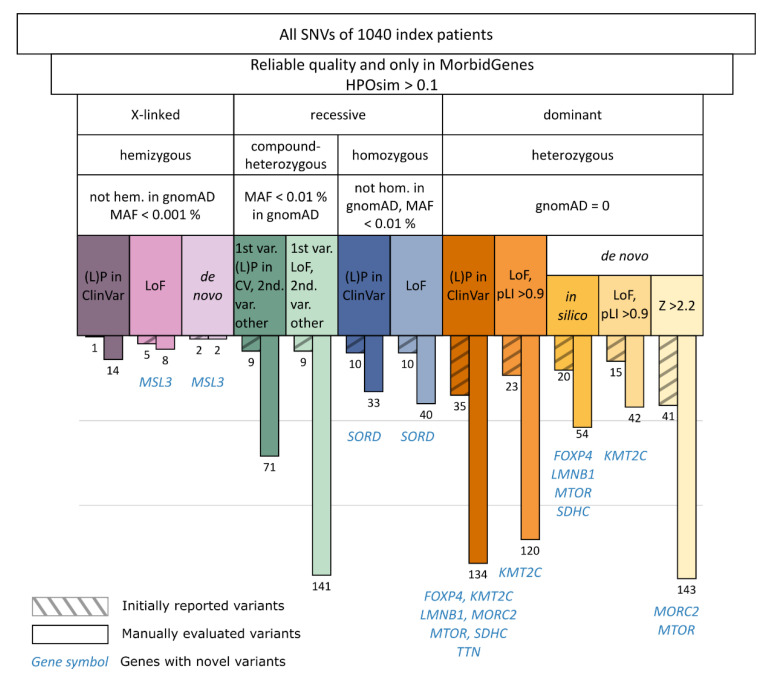
Filter cascade and resulting variants. The filter cascade consists of several steps that take into account the quality of the variant, the gene–disease association, and its phenotypic overlap with the symptoms of the corresponding individual. Depending on the zygosity, we filtered by the frequency of the variant in gnomAD and by other criteria, such as the variant effect, constraints, and presence in ClinVar, also considering segregation information (e.g., *de novo*). Numbers of manually evaluated variants are depicted for each arm of the filter cascade (bars without hatching). Initially reported (L)P variants that are covered by our filter cascade are displayed on the left in the colour of the corresponding filter criterion (bars with hatching). Notably, some variants appear in more than one filter step (e.g., a variant is both *de novo* and has been reported in ClinVar), and thus are counted repeatedly. In nine genes, we identified novel (L)P variants. CV: ClinVar, var.: variant, hem.: hemizygous, hom.: homozygous, LoF: loss of function, (L)P: (likely) pathogenic, MAF: minor allele frequency in gnomAD, pLI: probability of being loss-of-function-intolerant in gnomAD, Z: missense Z score in gnomAD.

**Figure 3 genes-14-00030-f003:**
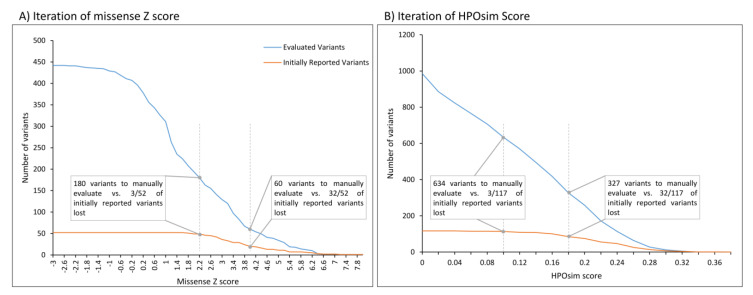
Iteration of missense Z score and HPOsim score. Number of variants to be manually evaluated (blue) and of initially reported (L)P variants (orange) based on missense Z score thresholds (**A**) and HPOsim score thresholds (**B**) as a single filter criterion. A) Depicted lines mark the adapted cut-off of 2.2 and the previously set cut-off of 4.0. B) Depicted lines mark the applied cut-off of 0.1 and an exemplary cut-off of 0.18. Notably, not all variants have an HPOSim score.

**Figure 4 genes-14-00030-f004:**
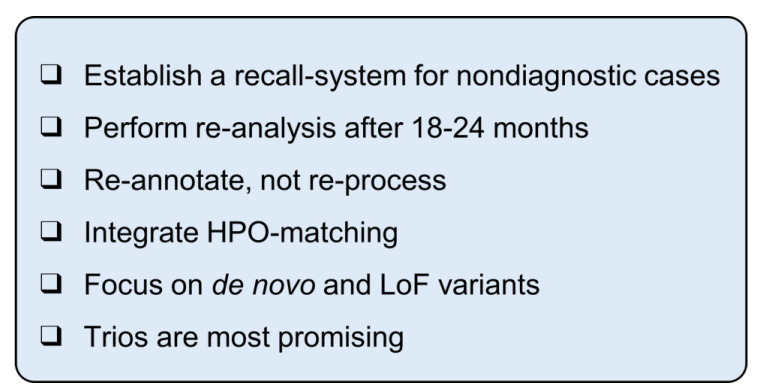
Checkbox for re-analysis of large cohorts.

**Table 1 genes-14-00030-t001:** Novel genetic diagnoses through re-analysis.

Gene	Approach (Date of Initial Evaluation)	Symptoms	Variant (Transcript, c-Code, p-Code)	Zygosity	Inheritance	ACMG Criteria (Final Classification)	OMIM Phenotype	Why Not Found Initially
**1*. FOXP4***	Trio (08/2017)	Hearing impairment, ventricular septal defect, flattened epiphysis, disproportionate short stature, craniofacial asymmetry	NM_001012426.2:c.1540 G > A, p.Ala514Thr	Heterozygous, *de novo*	Autosomal dominant	PS2, PS3, PS4_MOD, PM2_SUP, PP3 (pathogenic)	-	New gene–disease association after 53 months (4 years and 5 months)
**2*. KMT2C***	Trio (09/2017)	Hypothyroidism, mild intellectual disability, mild abnormality of facial shape, mild short stature	NM_170606.3:c.1829_ 1830delCA, p.Thr610Serfs * 4	Heterozygous*, de novo*	Autosomal dominant	PVS1, PS2_MOD, PS4_SUP, PM2_SUP (pathogenic)	Kleefstra syndrome 2 (#617768)	Updated caller, see also Appendix A
**3*. LMNB1***	Trio (12/2019)	Microcephaly, agenesis of the corpus callosum, cerebellar hypoplasia, growth retardation (prenatal)	NM_005573.4:c.97A > G, p.Lys33Glu	Heterozygous, *de novo*	Autosomal dominant	PS2_VSTR, PS3, PS4_MOD, PM2_SUP, PP3 (pathogenic)	Microcephaly 26, primary, autosomal dominant (#619179)	New gene–disease association after 9 months
**4*. MORC2***	Trio (05/2019)	Developmental delay, microcephaly	NM_001303256.3:c.79G > A, p.Glu27Lys	Heterozygous, *de novo*	Autosomal dominant	PS2_VSTR, PS3, PS4_MOD, PM2_SUP, PP2 (pathogenic)	Developmental delay, impaired growth, dysmorphic facies, and axonal neuropathy (#619090)	New gene–disease association after 15 months
**5*. MSL3***	Trio (01/2018)	Global developmental delay, seizures, chylothorax, mid-aortic syndrome	NM_078629.4:c.973_ 974delAG, p.Gln326Alafs * 5	Hemizygous, *de novo*	X-chromosomal dominant	PVS1, PS2, PS4_SUP, PM2_SUP (pathogenic)	Basilicata-Akhtar syndrome (#301032)	New gene–disease association after 9 months
**6. *MTOR***	Trio (9/2017)	Global developmental delay, macrocephaly	NM_004958.4:c.5911G > A, p.Ala1971Thr	Heterozygous, *de novo*	Autosomal dominant	PS2, PS4_SUP, PM2_SUP, PM5_SUP, PP2, PP3 (likely pathogenic)	Smith-Kingsmore syndrome (#616638)	Updated caller, see also Appendix A
**7*. SORD***	Trio (02/2020)	Pain in both legs from the age of 17, ataxia, atrophy of the leg muscles	NM_003104.5:c.757del, p.Ala253Glnfs * 27	Homozygous, maternal and paternal	Autosomal recessive	PVS1, PS3, PM3_VSTR (pathogenic)	Sorbitol dehydrogenase deficiency with peripheral neuropathy (#618912)	New gene–disease association after 3 months
**8*. SHDC ****	Trio (12/2017)	Mild global developmental delay, seizures, heterotopia, oral cleft, tall stature, obesity	NM_003001.5:c.377A > G, p.Tyr126Cys	Heterozygous, paternal	Autosomal dominant	PS4_MOD, PM1, PM2_SUP, PP3 (likely pathogenic)	Paragangliomas 3 (#605373)	Updated caller
**9*. TTN ****	Trio (09/2020)	Panhypopituitarism, developmental delay, patent ductus arteriosus, scoliosis, short stature, median cleft lip and palate	NM_001267550.2:c.80762_80765delAACA, p.Lys26921Argfs * 5	Heterozygous, maternal	Autosomal dominant	PVS1, PM2_SUP (likely pathogenic)	Cardiomyopathy, dilated, 1G (#604145)	Updated AMCG secondary findings list

Genes marked with * are on the ACMG secondary findings list (v3.0).

## Data Availability

Not applicable.

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
