# Peer review of "Approach to Cohort-Wide Re-Analysis of Exome Data in 1000 Individuals with Neurodevelopmental Disorders"

_genes, 2022, doi:10.3390/genes14010030_

Round 1

Reviewer 1 Report

Exome sequencing (ES) has a reported diagnostic yield between 30-50% in neurodevelopmental disordes. A “negative” exome in the remaining two-thirds of cases may be due to a number of factors. There are technical challenges such as imperfect sequencing, variant annotation, and filtering; together with difficulties in variant interpretation owing to incomplete knowledge of gene-/variant-disease associations, gene function; and the evolution of patient phenotype with time. One advantage of ES is the capacity for reanalysis of existing data in unsolved cases. 

Re-analysis of larger cohorts of non-diagnosed cases has been done before. The novelty of this paper, lies in the fact that authors not only re-analyze non-diagnostic cases but also diagnosed cases. The inclusion of diagnostic cases allowed them to examine the sensitivity of their filter cascade and to address recommendations on how to optimize them. In addition, authors use HPOsim scores which is a novel and exiting approach.

The paper is innovative, well-written and interesting to read. The topic is up-to-date. I only have a few suggestions for major and minor revisions.

Major revision:

Methods and materials:

1)      OMIM was used to evaluate the overlap between the variants and known phenotype. Since OMIM is not always updated, authors also used searched PubMed. This is a good approach but authors should describe which search words they use beside gene name to ensure capturing the clinical manuscripts that describe the human phenotypes.

2)      Authors describe in figure 1 that majority of patients had NDD and epilepsy. It´s however not clear to me what kind of epilepsy patients had. I understand it´s not a part of the scope of the paper but in order to improve transparency it´s important to know how many patients had early-onset and treatment  epilepsy

Results:

3)      It´s not clear for me why diagnostic yield was only 15%. Authors argue that this is because 699/1040 patients already underwent multi-gene panel and thus the low-hanging fruits were already detected and a clinical ES was only rarely able to detect a monogenic diagnosis. However, in 341/1040 no previous analysis was performed. We can presume that these patients are comparable with the 699 patients and that the majority have NDD/epilepsy. We can perhaps also presume that at least 20-30% of the 341 patient would have a monogenic disorder = 60-100 patients solved. Since 155/1040 reached a monogenic diagnoses, I´m struggeling to understand why the diagnostic yield was this low as I would expect that numbers would be higher especially due to 341 previously untested patients. I suggest authors explain and also in figure 1 show how many of the 699 and how many of the 341 patients were diagnosed with ES.

Discussion:

4)      Please discuss how you would suggest to set the MAF in countries with high number of consanguinity.

Minor revision:

Methods and materials:

1)      Authors should briefly describe number of reads and coverage in ES. Something in the line of “For all included subjects, the genomic regions targeted by the respective enrichment design had an average coverage of xxx reads, and >xx% were covered by at least xx reads.”

2)      The list of genes used was based on morbidgenes.org. Since this database is updated monthly, there should be a comment about this in the methods and discussion.

3)      Authors should offer a short description of what HPOsim score is, instead of just referring to File S1.

4)      Why were insilico prediction tools not applied to autosomal recessive and x linked disorders?

5)      Please explain which prediction tools were used to predict the effects of splicesite variants.

Results:

1)      The boxes within figure 3A and 3B have both an absolute number and a percentage. Why not add an absolute number in front of the four percentages?

Reviewer 2 Report

This is an interesting article in which the authors describe a re-analysis filter method to improve the diagnostic rate in patients studied by ES. It was achieved a 2.3% diagnostic rate in 1040 patients. For this re-analysis process, the authors suggest the application of specific criteria (Z-score, pLI and HPO-sim) to filter possible pathogenic variants while discussing its main limitations. Although I have some major/minor comments for the authors.

Major comments – figures 1 and 2 of the manuscript need to be reconsidered since they do not represent the paper content interpretation/discussion.

Materials and Methods – needs improvement

1.       Lines 58-59: It would be important to add int this first sentence the group of disorders those patients are affected, which is mainly NDD. Also, 7 cases are referred as other severe phenotypes, which phenotypes if not NDD?

2.       Figure 1 do not appropriately represent the paper content: line 62 referring as figure 1A, which is actually figure 1D. Line 68 referring as figure 1C which is actually figure 1A. Description is also not accurate since SureSelect Human All Exon V6 is from Agilent.

3.       Lines 75-76. Why CNV analysis were excluded since they are identified in Table_S1?

4.       Line 86: the authors in the autosomal dominant inheritance only refer to missense variants however other types of variants were also identified in their re-analysis (Table_S1)

5.       Line 79 authors use morbidgenes webpage (based on public databases  OMIM, PanelApp, SysNDD, ClinVar, HGMD and GenCC) to associate genes with a phenotype, line 108 the authors say using OMIM, HGMD or Pubmed. Can the authors clarify?

6.       From the 1040 patients how many were male/female, pediatric/adult?

7.       Figure 2: “2nd variant other” what does that mean in the context of compound heterozygous?  Which variants appear more than once in the filter step? Are those represented in the supplementary material?

Results – needs improvement

1.       Line 148: the remaining patients who do not have a NDD suspicion, had an unspecific phenotype?

2.       Lines 149-154 should be in the Materials and Methods section since figure 1 is displayed in this section. Also, part of the information in this sentence is repeated (lines 67-68)

3.       Line 156: please clarify since it is previously said CNV analysis were excluded.

4.       Lines 163-164: if those variants were published, PMID should be available in table_S2 along with the corresponding variant.

5.       Line 200: can the authors please clarify the several reasons variants were not captured by the filter cascade?

6.       Line 216:  “Out of these three, one was rescued by another filter criterion” which filter criterion?

7.       Line 219: “(1236 vs. 802 variants), while three initially reported variants were lost (see Figure 3 B)” this is not represented in figure 3B, please clarify.

Discussion – overall very well addressed describing the limitations detected in the re-analysis process and with general recommendations for genetic laboratories.

1.       Lines 250-256: bigger number of heterozygous variants makes sense since the NDD are usually associated with dominant autosomal inheritance. This information should be added since it supports the molecular results.

2.       Line 259: diagnostic yield of 15% not clear if it is about the first sequencing analysis? That means that initially the diagnosis rate achieved by ES was 15%? Because this is really low and not very common reported result.

3.       Why the authors have not considered to re-analyze variants classified initially as uncertain significance as a strategy to improve diagnostic rate? 

Reviewer 3 Report

The manuscript of Halfmeyer et al. in presenting a re-analysis pipeline for 1000 individuals with neurodevelopmental disorders, including trios and singletons.  Author describe precisely the newly diagnosed cases and discuss the pitfalls and advantages of the pipeline applied.

It would be important to add:

1) How much computational time did the re-analysis take

2) Who performed a manual variant review

Round 2

Reviewer 2 Report

The authors have provided all the responses and corrections required.